# Effect of tourniquet use on the risk of revision in total knee replacement surgery: an analysis of the National Joint Registry Data Set

Muhamed M Farhan-Alanie ![ORCID],[1,2] Yujin Lee,[1] Martin Underwood,[1] Andrew Metcalfe ![ORCID],[1,2] Mark J Wilkinson ![ORCID],[3] Andrew James Price,[4] Jane Warwick,[1] Peter David Henry Wall ![ORCID] [1,2]

[1]Clinical Trials Unit, Warwick Medical School, University of Warwick, Coventry, UK
[2]Trauma and Orthopaedics, University Hospitals Coventry and Warwickshire NHS Trust, Coventry, UK
[3]Department of Oncology and Metabolism, The University of Sheffield, Sheffield, UK
[4]Nuffield Department of Orthopaedics, Rheumatology and Musculoskeletal Sciences, University of Oxford, Oxford, UK

**Correspondence to**
Muhamed M Farhan-Alanie; muhamed.farhan-alanie@nhs.net

## ABSTRACT

**Objective** Tourniquet use in total knee replacement (TKR) is believed to improve the bone-cement interface by reducing bleeding, potentially prolonging implant survival. This study aimed to compare the risk of revision for primary cemented TKR performed with or without a tourniquet.

**Design** We analysed data from the National Joint Registry (NJR) for all primary cemented TKRs performed in England and Wales between April 2003 and December 2003. Kaplan-Meier plots and Cox regression were used to assess the influence of tourniquet use, age at time of surgery, sex and American Society of Anaesthesiologists (ASA) classification on risk of revision for all-causes.

**Results** Data were available for 16 974 cases of primary cemented TKR, of which 16 132 had surgery with a tourniquet and 842 had surgery without a tourniquet. At 10 years, 3.8% had undergone revision (95% CI 2.6% to 5.5%) in the no-tourniquet group and 3.1% in the tourniquet group (95% CI 2.8% to 3.4%). After adjusting for age at primary surgery, gender and primary ASA score, the HR for all-cause revision for cemented TKR without a tourniquet was 0.82 (95% CI 0.57 to 1.18).

**Conclusions** We did not find evidence that using a tourniquet for primary cemented TKR offers a clinically important or statistically significant reduction in the risk of all-cause revision up to 13 years after surgery. Surgeons should consider this evidence when deciding whether to use a tourniquet for cemented TKR.

## BACKGROUND

Total knee replacement (TKR) is widely used to relieve pain and improve function for individuals with end-stage symptomatic arthritis of the knee.[1 2] In the UK, the majority of TKR components are cemented in place to hold and stabilise them in the correct position in the bone.[3] The surgery can be performed with or without the use of a thigh tourniquet. Survey research has identified that most surgeons in the UK and around the world perform TKR with the aid of a tourniquet and that this practice has remained unchanged over time.[4–6] In 2003, 93% of primary TKRs were performed with a tourniquet. A 2015 survey of the British Association of Knee Surgeons (BASK) indicated that 90% of surgeons routinely use a tourniquet.[7 8] The reasons for the widespread use of a tourniquet include a belief that bleeding bone surfaces might impair the fixation of cemented prostheses and reduce long-term implant survivorship.[7] There are data from laboratory animal studies that suggest blood interferes with the strength of the bone-cement interface.[9] This may translate to an increased risk of aseptic loosening of the implant components, the most common cause of TKR failure necessitating revision surgery.[3] However, there is also evidence suggesting that tourniquets do not improve implant fixation and longevity. Two randomised controlled trials (RCTs) in humans which used radiostereometric analysis (RSA — a surrogate marker of long-term

### Strengths and limitations of this study

► This is the largest and longest cohort comparing revision for all-causes following total knee replacement (TKR) surgery performed with versus without a tourniquet.
► Approximately 17 000 TKR procedures from 384 centres were included in this study, enabling generalisability of our results to similar patient groups.
► We have considered all-cause revision as our primary outcome to mitigate the risk of misclassification of indication for revision.
► Patients were followed up for a median of 12.2 years from their primary procedure, and we have accounted for patient death in our analysis.
► We could not control for confounding factors including tourniquet inflation timing, pressure and duration as these details were not collected in the National Joint Registry minimum dataset and patients were not randomised.

BMJ

implant survival) demonstrated no significant differences in tibial component migration between TKRs performed with versus without tourniquet at up to two years of follow-up.[10 11]

Tourniquet use is not without its risks and has been associated with various adverse effects such as increased pain and venous thromboembolism.[12] They have also been associated with other complications including haematoma formation, wound discharge, skin injuries and nerve palsy, as well as reduced knee range of motion, even two years after surgery.[4 10] The routine use of tourniquets in TKR surgery therefore requires scrutiny to determine whether they help to reduce the risk of revision surgery or merely expose patients to unnecessary harm.[12] This study compares the all-cause revision rate between primary cemented TKR performed with or without tourniquet. If tourniquet use is effective in improving the bone-cement interface then lower rates of revision would be observed in the tourniquet group.

## METHODS

This work forms part of a larger research project titled Safety and Feasibility Evaluation of Tourniquets in Knee Replacement Surgery (SAFE-TKR) and adhered to the published protocol.[13] This article reports the findings for risk of revision between patients who underwent TKR surgery with versus without a thigh tourniquet. Other outcomes examining potential adverse events and length of hospital will be reported within a separate article.

Data covering England and Wales were provided by the NJR for the period in which tourniquet use was recorded as part of the minimum dataset for knee replacement surgery (April 2003–December 2003).[3 8] We analysed only data from primary elective cemented TKRs of which the vast majority were performed for osteoarthritis.[8] Tourniquet use, along with other routinely collected NJR baseline variables including age at time of surgery, sex and American Society of Anaesthesiologists (ASA) classification, was analysed to measure independent effects, if any, on the rate and timing of all-cause revision.

Kaplan-Meier plots were used to compare time to revision between groups which underwent TKR with and without a tourniquet. Cox regression was performed to quantify the difference between the two groups using hazard ratios (HR) and to assess the effect of the selected baseline variables on risk of all-cause revision. Time to revision was measured from date of index surgery to the earlier of revision and either date of last follow-up or death. All outcomes other than revision were censored. Statistical analyses was performed using STATA V.14 Data and Analysis Software.

### Patient and public involvement

Thirty patients who have undergone TKR were surveyed and took part in a focus group to explore their experiences and views towards tourniquet use during their surgery. Results revealed that patients believed further research on tourniquet use in TKR was important. Implant longevity was identified as being a particular concern among the patients consulted. Patients contributed to the design of this study by helping determine the important outcomes and their timing. The results of this research will be disseminated to study participants through press releases and social media.

## RESULTS

We analysed 20 479 primary knee replacement surgery cases done between April 2003 and December 2003. Of the 406 hospitals listed within the NJR system, 384 had returned data (94%).[5] The numbers undergoing each type of knee replacement surgery, by tourniquet use, are given in table 1.

Our analysis is of the 16 974 cases of primary cemented TKR. Median follow-up was 12.2 years (IQR 8.4–12.6). The baseline characteristics and outcome variables for this subset are shown in tables 2 and 3, respectively.

HRs and associated 95% CIs from the univariable and multivariable Cox regression models are shown in table 4. In the univariable model, there was no evidence that the rate at which revisions occur over the study period of 13.3 years differs between those undergoing surgery with a tourniquet and those undergoing surgery without (HR=0.83, 95% CI 0.57 to 1.20, p=0.31). Of the selected variables considered, only age at primary surgery and gender were significant in the univariable models. After adjusting for age at primary surgery, gender and

| Table 1 | Types of knee replacement surgery in the England and Wales NJR dataset (April 2003–December 2003) | | |
|---|---|---|---|
| | **Tourniquet used (n=19 330)** | **Tourniquet not used (n=1149)** | **Total (N=20 479)** |
| Type of replacement implant, n (%) | | | |
| TKR cemented | 16 132 (83) | 842 (73) | 16 974 (83) |
| TKR uncemented | 1238 (6) | 228 (20) | 1466 (7) |
| TKR unclassified (eg, hybrid) | 272 (1) | 12 (1) | 284 (1) |
| Unicondylar | 1491 (8) | 57 (5) | 1548 (8) |
| Patellofemoral | 197 (1) | 10 (1) | 207 (1) |

NJR, National Joint Registry; TKR, total knee replacement.

**Table 2** Baseline characteristics for all primary cemented TKRs

| | Tourniquet used (n=16 132) | Tourniquet not used (n=842) | Total (N=16 974) |
|---|---|---|---|
| Age at primary surgery (years), mean (SD) | 71 (9) | 72 (9) | 71 (9) |
| Patient gender, n (%) | | | |
| Male | 6871 (43) | 382 (45) | 7253 (43) |
| Female | 9261 (57) | 460 (55) | 9721 (57) |
| Primary ASA score, n (%) | | | |
| P1: fit and healthy | 4421 (28) | 235 (28) | 4656 (28) |
| P2: mild disease not incapacitating | 9848 (61) | 508 (60) | 10 356 (61) |
| P3: incapacitating systemic disease | 1828 (11) | 97 (12) | 1925 (11) |
| P4: life-threatening disease | 35 (<1) | 2 (<1) | 37 (<1) |

ASA, American Society of Anaesthesiologists; TKR, total knee replacement.

primary ASA grade in the multivariable model, the effect of tourniquet use on the rate at which revisions occur remained largely unchanged (HR=0.82, 95% CI 0.57 to 1.18, p=0.29). Age at primary surgery remained highly significant in the multivariable model and the HRs did not change (global test, p<0.001), suggesting that age is both a strong and independent predictor of risk of all-cause revision. Gender and primary ASA grade did not reach statistical significance in the multivariable model, suggesting these were not reliable independent predictors for risk of all-cause revision. Figure 1 shows Kaplan-Meier survival curves for time to all-cause revision in patients having TKR surgery with and without a tourniquet.

The difference in percentage unrevised over time (cemented TKR with tourniquet minus cemented TKR without a tourniquet) is presented, together with pointwise confidence intervals, in figure 2. At 5 years, 2.1% had undergone revision (95% CI 1.3% to 3.3%) in the no-tourniquet group and 1.7% in the tourniquet group (95% CI 1.49% to 1.9%). At 10 years, 3.8% had undergone revision (95% CI 2.6% to 5.5%) in the no-tourniquet group compared with 3.1% in the tourniquet group (95% CI 2.8% to 3.4%). Reference lines (indicated red

in figure 2) indicate where the magnitude of this difference exceeds 1%. The point estimates of the difference (solid blue line) lie within these two limits throughout. The upper end of the confidence interval does, however, exceed this limit from approximately 3 years onwards. Thus, a difference of more than 1% in either direction between the tourniquet and no-tourniquet groups cannot be ruled out.

## DISCUSSION

This analysis used data from the world's largest TKR audit dataset, the NJR, and enabled us to examine the association of tourniquet use with all-cause revision surgery following primary, elective cemented TKR. Our results show similar revision rates between people undergoing primary TKR with versus without tourniquet up to 13.3 years after surgery. The only baseline variable studied that showed evidence of an independent effect on the risk of revision was age at primary surgery, consistent with previous research.[14] We also observed that a tourniquet was used in 95% of cemented TKRs and 84% of cementless TKRs. From this, we infer that tourniquet use might

**Table 3** Outcome variables for all primary cemented TKRs

| | Tourniquet used (n=16 132) | Tourniquet not used (n=842) | Total (N=16 974) |
|---|---|---|---|
| Outcome, n (%) | | | |
| Unrevised | 9490 (59) | 466 (55) | 9956 (59) |
| Revised | 493 (3) | 30 (4) | 523 (3) |
| Unrevised at time of death | 6149 (38) | 346 (41) | 6495 (38) |
| Time to outcome in years, median (IQR) | 12.2 (8.4–12.6) | 12.2 (7.5–12.6) | 12.2 (8.4–12.6) |
| Timing of death, n (%) | | | |
| Within 30 days | 59 (<1) | 3 (<1) | 62 (<1) |
| 31–90 days | 42 (<1) | 2 (<1) | 44 (<1) |
| After 90 days | 6048 (37) | 341 (40) | 6389 (38) |

IQR, interquartile range; TKR, total knee replacement.

**Table 4** HRs and associated 95% CIs from the univariable and multivariable Cox regression models for all-cause revision

| | Univariable models | | Multivariable model | |
|---|---|---|---|---|
| | HR (95% CI) | P value | HR (95% CI) | P value |
| **Tourniquet used** | | | | |
| No | 1.00 | | 1.00 | |
| Yes | 0.83 (0.57 to 1.20) | 0.31 | 0.82 (0.57 to 1.18) | 0.29 |
| **Age at primary surgery (years)** | | | 0.17 (0.11 to 0.27) | |
| <65 | 1.00 | | 1.00 | |
| 65–69 | 0.75 (0.60 to 0.94) | | 0.74 (0.60 to 0.93) | |
| 70–79 | 0.41 (0.33 to 0.50) | | 0.40 (0.32 to 0.49) | |
| >80 | 0.18 (0.11 to 0.27) | <0.001 | 0.17 (0.11 to 0.27) | <0.001 |
| **Patient gender** | | | | |
| Female | 1.00 | | 1.0 | |
| Male | 1.21 (1.01 to 1.43) | 0.03 | 1.12 (0.94 to 1.33) | 0.21 |
| **Primary ASA score, n (%)** | | | | |
| P1 | 1.00 | | 1.00 | |
| P2 | 0.90 (0.74 to 1.10) | | 1.08 (0.89 to 1.31) | |
| P3 | 1.13 (0.84 to 1.51) | | 1.39 (1.04 to 1.87) | |
| P4 | 1.34 (0.19 to 9.54) | 0.39 | 1.61 (0.22 to 11.48) | 0.19 |

ASA, American Society of Anaesthesiologists.

be greater in cemented procedures because of a perceived association with better cementation quality. However, this association may have arisen by chance.

Our findings indicate that performing TKR with a tourniquet is not associated with any clinically meaningful difference in the risk of revision compared with surgery without a tourniquet. These findings are in keeping with two RCTs of 50 and 60 patients who underwent primary cemented TKR surgery for osteoarthritis with or without the use of a tourniquet. No significant differences in tibial component migration was observed at up to 2 years of follow-up using radiostereometric analysis.[10 11] However, patient loss to follow-up in both RCTs resulted in inadequate power to detect differences in migration following

one year, limiting the validity of the study findings to predict future revision for loosening to between five and ten prosthesis years.[15] In contrast, our study has compared all-cause revision up to 13.3 years postoperatively and is more reflective of real-life practice as it does not limit surgeons to a defined surgical technique or other factors such as choice of implant or bone cement.

A Cochrane systematic review has shown that using a tourniquet for TKR surgery is associated with an increased risk of serious adverse events which includes venous thromboembolism, infection, reoperation and mortality. Other complications including blisters,

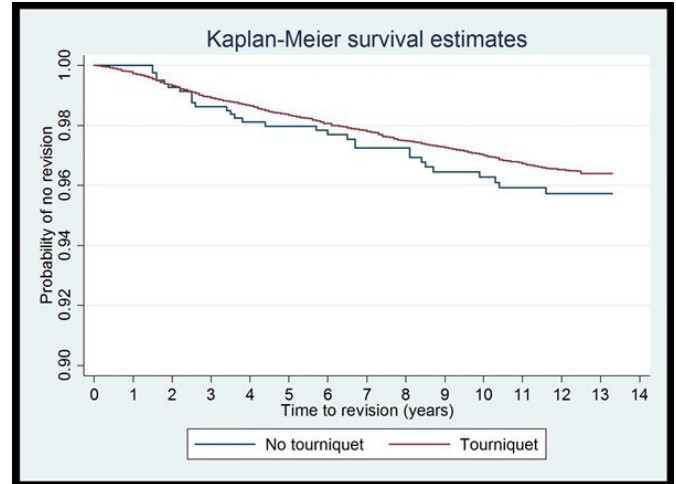

**Figure 1** Time to all-cause revision by tourniquet use.

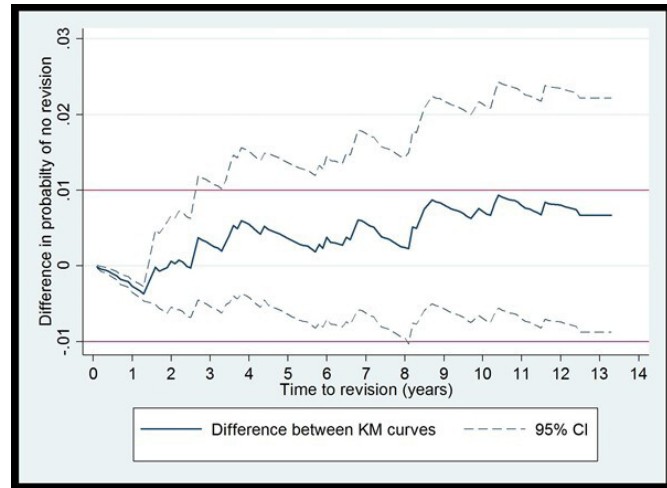

**Figure 2** Difference in time to revision (TKR with tourniquet – TKR without tourniquet), over time (unadjusted). TKR, total knee replacement.

haematoma, wound oozing, bruising and nerve palsy have also been shown to be more common with tourniquet use during TKR surgery, and the devices themselves are frequently colonised with bacteria.[4 16 17] It has been shown that tourniquet use is associated with a modest statistically significant reduction in intraoperative blood loss. This may improve the surgical field of view, but we are not aware of any published literature formally examining this outcome. It is surprising that in the absence of any high-quality evidence demonstrating any likely benefits from using a tourniquet and the existing evidence demonstrating their increased risk of harm to patients, they continue to have such widespread use. Furthermore, there may also be a risk of systemic emboli occurring following the deflation of a tourniquet.[18] Transcranial Doppler ultrasound studies have found echogenic material in the Circle of Willis after a tourniquet is released and that micro-embolism can occur even in the absence of a patent foramen ovale.[19] The potential benefits of tourniquet use during TKR surgery must therefore be balanced against the risks.

We analysed approximately 17 000 TKR procedures from over 384 centres which represent 94% of hospitals listed in the NJR at that time.[8] Patients were also followed up for a median of 12.2 years from their primary procedure, enabling any potential differences in revision rates to become more apparent particularly as risk of revision for aseptic loosening increases over time.[3] We have used all-cause revision as the endpoint in our study to mitigate risk of misclassification of aseptic loosening within the NJR. This is a global outcome that considers other failure mechanisms responsible for revision procedure, such as prosthetic joint infection, especially considering that wound complications have been associated with tourniquet usage.[4 20] This association is believed to be due to a combination of factors including wound ischaemia and local inflammation.[21 22] The relatively short duration in which tourniquet data was captured within the NJR minimum dataset (April–December 2003) reduces biases by time-dependent unknown variables, such as cementation techniques and polyethylene modification, but does limit the sample size. Conversely, the minimum dataset has evolved over time to include established risk factors for revision such as Body Mass Index which were not captured within version 1 and therefore could not be included in our regression analysis.[23 24] Further limitations also related to application of registry-based data include missing data and under-reporting of procedures, preventing the linkage of revision procedures to any index procedure.[3 25] Due to missing data in our dataset, we were not able to perform a subanalysis of revision by indication and assess whether rates of periprosthetic patella fractures and secondary patellar resurfacing differed between the two groups. We acknowledge that we did not consider surgery-related factors such as implant brand, constraint and bearing type in our analysis. This was due to very limited numbers in these subgroups which would not add further meaning to our analysis. Cement brand and viscosity were also not considered, but antibiotic content has not been shown to affect all-cause revision.[26] However, we have collected data that encompass the diversity of implant designs and cement characteristics across routine practice in the UK, and this helps ensure wide generalisability of the results. Another limitation is the relatively small sample size of TKR procedures performed without a tourniquet across a large number of centres (approximately 150 centres) which may have influenced our results due to performance bias and confounding by indication.

Due to the defined minimum dataset, we could not account for other factors such as use of tranexamic acid and application of hypotensive anaesthesia, which are used to limit bleeding from cancellous bone ends in an attempt to improve the bone-cement interface. Also, variability in tourniquet inflation timing, pressures and duration introduces heterogeneity within the tourniquet group with some surgeons maintaining inflation throughout the entire procedure and others only during the cementation process. However, the study is reflective of real-life routine. Lastly, there is also a risk of bias in our cohort study due to confounding by indication although an RCT to address this is not feasible given the number of participants required to capture small differences in implant survival and the extended follow-up required. Despite the majority of participants (95%) having surgery with a tourniquet, there were a large number in the no-tourniquet group, and the baseline patient characteristics between the two groups are broadly similar.

## CONCLUSIONS

Our study has shown a similar risk of revision for primary cemented TKRs performed with and without a tourniquet. However, we are unable to exclude conclusively a clinically important improvement in long-term survival for cemented TKRs undertaken with a tourniquet. Surgeons should carefully consider this study's findings as well as the known risks and benefits of tourniquet use, and discuss these issues during the consent process to allow patients to make an informed decision.

**Acknowledgements** We thank the patients and staff of all the hospitals who have contributed data to the National Joint Registry (NJR). We are grateful to the Healthcare Quality Improvement Partnership, the NJR Steering Committee and staff at the NJR for facilitating this work. We also thank all patients and public members/representatives who were involved in this study and, in particular, CG, JS and JD for their ongoing contribution throughout the study.

**Contributors** MMF-A: drafted and reviewed the final manuscript. MU, AM, MJW and AJP: study design, drafted and reviewed the final manuscript. YL and JW: data analysis, drafted and reviewed the final manuscript. PDHW: study conception, study design, data analysis, drafted and reviewed the final manuscript.

**Funding** This study presents research funded by a National Institute for Health Research (NIHR) Post-Doctoral Fellowship Award (PDF-2015-08-108).

**Competing interests** None declared.

**Patient consent for publication** Not required.

**Ethics approval**  The project was approved by the National Joint Registry (NJR) Research Committee (May 2016) and the National Research Ethics Committee (January 2016) (15/WM/0455). Patient consent was obtained for data collection by the NJR. According to the specifications of the NHS Health Research Authority, separate informed consent and ethical approval were not required for the present study.

**Provenance and peer review**  Not commissioned; externally peer reviewed.

**Data availability statement**  Access to the data analysed in this study required permission from the National Joint Registry for England, Wales and Northern Ireland Research Sub-committee. http://www.njrcentre.org.uk/njrcentre/Research/Researchrequests/tabid/305/Default.aspx contains information on research data access request to the National Joint Registry.

**ORCID iDs**
Muhamed M Farhan-Alanie http://orcid.org/0000-0002-9209-0108
Andrew Metcalfe http://orcid.org/0000-0002-4515-8202
Mark J Wilkinson http://orcid.org/0000-0001-5577-3674
Peter David Henry Wall http://orcid.org/0000-0003-3149-3373

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
