## [Reviewer comments · BMJ Open]

ARTICLE DETAILS

TITLE (PROVISIONAL)	The effect of tourniquet use on the risk of revision in total knee replacement surgery: an analysis of the National Joint Registry dataset
AUTHORS	Farhan-Alanie, Muhamed; Lee, Yujin; Underwood, Martin; Metcalfe, Andrew; Wilkinson, J. Mark; Price, Andrew; Warwick, Jane; Wall, Peter

VERSION 1 – REVIEW

REVIEWER	Sezgin , Erdem A Aksaray University, Orthopedics and Traumatology
REVIEW RETURNED	17-Jan-2021

GENERAL COMMENTS	The study “The effect of tourniquet use on the risk of revision in total knee replacement surgery: an analysis of the National Joint Registry dataset” is based on the largest dataset with longest follow-up on the topic. Manuscript is well-written on a topic that is being questioned by many arthroplasty surgeons, although had been only investigated in a small number of scientific studies limited by their sample size or follow-up duration. The study is excellently presented, and the collected data appear to be solid. The analyses are relevant. The discussion is comprehensive and highlights the uniqueness of the data harvested. Limitations are also well described which enlightens the route for future research. I have a few remarks. 1- Study protocol had been published earlier, as cited by the authors. However, although the protocol states risk of early adverse events such as venous thromboembolism, cerebrovascular accidents and length of stay would be compared between groups; those analyses were not emphasized in the study. I agree that the relevant outcome measure is long-term revision rates however I think authors could have discussed why they chose to omit these data, like they discussed omitting the data on types of implants. 2- In the 2004 NJR annual report, cited by the authors, 38.8% of primary total condylar knee replacements included patella resurfacing. Revision rates can be affected by patella resurfacing; some being due to addition of a patella resurfacing in a second surgery (classified as a revision) and some due to complications of the primary resurfacing (e.g., implant failure, fracture). Thus, I believe including or briefly mentioning whether patellar resurfacing differs between two groups, would allow a more accurate assessment.
---

	3- I agree that using all-cause revision as an end point helps to mitigate the risk of misclassification of aseptic loosening. It also considers the potentially increased rates of revision due to expected increased wound complications in the tourniquet group. However, as procedures such as secondary patella resurfacing, exchange of insert, DAIR, first stage of a two-stage revision may have been classified as revisions, removing those from the data which would have increased the accuracy of the analysis for loosening without increasing the risk of misclassification. I think definition of revision can be further detailed in the manuscript. I also suggest presenting indications and techniques for revisions without including them in the analyses (as there are only 30 all-cause revisions in the second group). This would help inform and guide the readers to agree with the authors that analyzing all-cause revision was valid in a study that was primarily aimed to analyze implant survivorship based on potential problems in the bone-cement interface. 4- I acknowledge that this study compares two large series (16,132 vs 842). However, as no-tourniquet group consisted of about 5% of the study group, I suggest considering mentioning whether a specific center or group of surgeons adopted no-tourniquet strategy. As clinical outcomes may differ between centers, a point referring to this problem in the discussion section could be added. Although this would be considered as another limitation, if only a small group of surgeons/centers had adopted this technique in their routines, effect of potential bias due to indication may be mitigated. I understand with such a large dataset there will always be a variance and authors would not be able to reliably adjust for all these factors. Thus, I agree reflecting the real-life routine is completely acceptable. However, I think aforementioned potential limitation should be discussed. 5- In Table 3, 346 patients were unrevised at the time of death. But 449 were reported to be dead during follow up. As only 346 of them were unrevised at the time of death, if I am interpreting the data correctly, there should have been 103 patients who were dead, but revised. Whereas there are 30 revisions. Can there be an error in this table? 6- Number of participants (TKA-Cemented) was 16974, and 16132 had surgery with a tourniquet so I suggest correcting the rate 94.4% mentioned in discussion to 95% (16132/16974).
--	---

REVIEWER	Mikkonen, Santtu University of Eastern Finland
REVIEW RETURNED	17-Jan-2021

GENERAL COMMENTS	This review concentrates only on statistical methodology used in the study The Cox mode applied is rather simple, and thus need no thorough revision. I only have some concerns on presenting the results: Why interpreting univariable models in detail? These data clearly require multivariable approach, as the model needs to be adjusted for age, sex and ASA classification before drawing any conclusions. Decrease the interpretation of univariable model and discuss more the multivariable solution.
--

	Specific comment: Page 8, row 6: the numbers are given in Table 4, why they are repeated here? this decreases readability of the text
--	--

REVIEWER	Turcotte, Justine J Anne Arundel Medical Center Orthopedics
REVIEW RETURNED	19-Jan-2021

GENERAL COMMENTS	Thank you for the opportunity to review this work entitled, “The effect of tourniquet use on the risk of revision in total knee replacement surgery: an analysis of the National Joint Registry dataset”. Using a cohort of approximately 17,000 TKRs with a median of 12.2 years of follow up, the authors have evaluated the impact of tourniquet use on all-cause revision rates following TKR. Based upon both univariate and multivariate methods, tourniquet use was not associated with decreased risk of revision. However, evaluation of the results over time does trend toward increased probability of no-revision after 3 years in the tourniquet group. The authors should be commended for using a large national dataset to address an important question that is not easily answered due to the need for long-term follow up. The methods and results are well described, and the manuscript is well written. I recommend the following minor revisions to the work. Pg. 3, Line 55 – Reword this sentence. Tourniquets do not necessarily harm patients. Consider changing the end of this sentence to “...or merely expose patients to increased risk of complications.” Pg. 7, Line 51 – I recommend moving your sentence that begins “Figure 1 depicts...” down to the last paragraph of results beginning on pg. 8 line 38. It will flow better to introduce the KM curve and then trends in differences at the various time points. Pg. 8 – Please add to the first paragraph to state that gender was no longer a significant risk factor for revision after adjusting for other factors in the multivariate model. Pg. 10, paragraph beginning on line 30 – Please provide an expanded summary of the findings and limitations from these two trials. Since these are the best historical foundations for the current study, more detail and comparison with your results is warranted. Pg. 10, paragraph beginning on line 44 – Please modify this paragraph to present a more balanced description of why tourniquet use remains so prevalent. Please cite the purported benefits of increased intraoperative visualization, and that intraoperative blood loss results are mixed, as multiple studies have found reduced blood loss. Pg. 11 -12 – The limitations of this study are appropriately described. I look forward to reading a follow up to this study using the expanded data collected in updated versions of the NJR dataset.
---

VERSION 1 – AUTHOR RESPONSE

Authors' response

Thank you for highlighting this. We have revised our manuscript to include this requested information.

Reviewer: 1

Dr. Erdem A Sezgin , Aksaray University

Comments to the Author:

The study “The effect of tourniquet use on the risk of revision in total knee replacement surgery: an analysis of the National Joint Registry dataset” is based on the largest dataset with longest follow-up on the topic. Manuscript is well-written on a topic that is being questioned by many arthroplasty surgeons, although had been only investigated in a small number of scientific studies limited by their sample size or follow-up duration. The study is excellently presented, and the collected data appear to be solid. The analyses are relevant. The discussion is comprehensive and highlights the uniqueness of the data harvested. Limitations are also well described which enlightens the route for future research.

I have a few remarks.

Comment 1

Study protocol had been published earlier, as cited by the authors. However, although the protocol states risk of early adverse events such as venous thromboembolism, cerebrovascular accidents and length of stay would be compared between groups; those analyses were not emphasized in the study. I agree that the relevant outcome measure is long-term revision rates however I think authors could have discussed why they chose to omit these data, like they discussed omitting the data on types of implants.

Authors' response

The results of these outcomes were not included in the paper as our focus was on the risk of revision between groups and we did not want to take the emphasis away from this. We have added a statement to make this clear in our manuscript. The other outcomes will be reported as a separate research paper focussing on adverse events.

Comment 2

In the 2004 NJR annual report, cited by the authors, 38.8% of primary total condylar knee replacements included patella resurfacing. Revision rates can be affected by patella resurfacing; some being due to addition of a patella resurfacing in a second surgery (classified as a revision) and some due to complications of the primary resurfacing (e.g., implant failure, fracture). Thus, I believe including or briefly mentioning whether patellar resurfacing differs between two groups, would allow a more accurate assessment.

Authors' response

Thank you for highlighting this issue. Unfortunately, data regarding revision for these indications was not available (missing) for the majority of procedures. We appreciate this is a limitation of our study and have now included this point in the discussion.

Comment 3

I agree that using all-cause revision as an end point helps to mitigate the risk of misclassification of aseptic loosening. It also considers the potentially increased rates of revision due to expected increased wound complications in the tourniquet group. However, as procedures such as secondary patella resurfacing, exchange of insert, DAIR, first stage of a two-stage revision may have been classified as revisions, removing those from the data which would have increased the accuracy of the analysis for loosening without increasing the risk of misclassification. I think definition of revision can be further detailed in the manuscript. I also suggest presenting indications and techniques for revisions without including them in the analyses (as there are only 30 all-cause revisions in the second group). This would help inform and guide the readers to agree with the authors that analyzing all-cause revision was valid in a study that was primarily aimed to analyze implant survivorship based on potential problems in the bone-cement interface.

Authors' response

Thank you for this comment. The dataset did not provide sufficient detail to enable exclusion of procedures such as secondary patella resurfacing, exchange of insert, and DAIR. First and second stage revision procedures were included in our all-cause revision analysis however these were not double counted. These procedures were included as we defined our primary outcome as revision for all-causes from the outset of the study (as per published protocol (1)) to help mitigate the risk of misclassification of the indication for revision. Differentiating between aseptic loosening and infection can be difficult – studies have shown that the incidence of culture-negative but suspected periprosthetic joint infection is 15% and these patients may have undergone a staged revision procedure recorded as being performed for infection despite uncertainty of the diagnosis however may actually suffer from aseptic loosening (2). Conversely, many revisions are recorded as being performed for aseptic loosening at the time of the procedure however tissue samples sent for culture return back positive several days post-operatively.

Comment 4

I acknowledge that this study compares two large series (16,132 vs 842). However, as no-tourniquet group consisted of about 5% of the study group, I suggest considering mentioning whether a specific center or group of surgeons adopted no-tourniquet strategy. As clinical outcomes may differ between centers, a point referring to this problem in the discussion section could be added. Although this would be considered as another limitation, if only a small group of surgeons/centers had adopted this technique in their routines, effect of potential bias due to indication may be mitigated. I understand with such a large dataset there will always be a variance and authors would not be able to reliably adjust for all these factors. Thus, I agree reflecting the real-life routine is completely acceptable. However, I think aforementioned potential limitation should be discussed.

Authors' response

Thank you for raising this point. We have now included a statement in our limitations paragraph to discuss the issues highlighted.

Comment 5

In Table 3, 346 patients were unrevised at the time of death. But 449 were reported to be dead during follow up. As only 346 of them were unrevised at the time of death, if I am interpreting the data correctly, there should have been 103 patients who were dead, but revised. Whereas there are 30

revisions. Can there be an error in this table?

Authors' response

Thank you identifying this error. The data in this part of the table included results for all primary TKR procedures and has now been corrected to reflect cemented TKR procedures only.

Comment 6

Number of participants (TKA-Cemented) was 16974, and 16132 had surgery with a tourniquet so I suggest correcting the rate 94.4% mentioned in discussion to 95% (16132/16974).

Authors' response

Thank you for noticing this error which we have corrected.

Reviewer: 2

Dr. Santtu Mikkonen, University of Eastern Finland

Comments to the Author:

This review concentrates only on statistical methodology used in the study

The Cox mode applied is rather simple, and thus need no thorough revision. I only have some concerns on presenting the results:

Why interpreting univariable models in detail? These data clearly require multivariable approach, as the model needs to be adjusted for age, sex and ASA classification before drawing any conclusions. Decrease the interpretation of univariable model and discuss more the multivariable solution.

Authors' response

Thank you for this comment. We have cut back on the text describing the results of the univariable model and included further text regarding the results of the multivariable model.

Comment 1

Page 8, row 6: the numbers are given in Table 4, why they are repeated here? this decreases readability of the text

Authors' response

Thank you for this perspective. We have removed the results that are most likely to affect readability of the text.

Reviewer: 3

Dr. Justine J Turcotte, Anne Arundel Medical Center Orthopedics

Comments to the Author:

Thank you for the opportunity to review this work entitled, "The effect of tourniquet use on the risk of revision in total knee replacement surgery: an analysis of the National Joint Registry dataset". Using a

cohort of approximately 17,000 TKRs with a median of 12.2 years of follow up, the authors have evaluated the impact of tourniquet use on all-cause revision rates following TKR. Based upon both univariate and multivariate methods, tourniquet use was not associated with decreased risk of revision. However, evaluation of the results over time does trend toward increased probability of no-revision after 3 years in the tourniquet group.

The authors should be commended for using a large national dataset to address an important question that is not easily answered due to the need for long-term follow up. The methods and results are well described, and the manuscript is well written. I recommend the following minor revisions to the work.

Comment 1

Pg. 3, Line 55 – Reword this sentence. Tourniquets do not necessarily harm patients. Consider changing the end of this sentence to “...or merely expose patients to increased risk of complications.”

Authors' response

We have now supported this statement by referencing a Cochrane review on the use of tourniquets in TKR surgery. Results of this meta-analysis showed a statistically significant difference in multiple outcomes including pain, serious adverse events, venous thromboembolism and infection.

Comment 2

Pg. 7, Line 51 – I recommend moving your sentence that begins “Figure 1 depicts...” down to the last paragraph of results beginning on pg. 8 line 38. It will flow better to introduce the KM curve and then trends in differences at the various time points.

Authors' response

Thank you for this suggestion. We have now moved this sentence from the start to the end of the paragraph.

Comment 3

Pg. 8 – Please add to the first paragraph to state that gender was no longer a significant risk factor for revision after adjusting for other factors in the multivariate model.

Authors' response

Thank you. We have now explicitly reported this in the text.

Comment 4

Pg. 10, paragraph beginning on line 30 – Please provide an expanded summary of the findings and limitations from these two trials. Since these are the best historical foundations for the current study, more detail and comparison with your results is warranted.

Authors' response

Thank you for this comment. We have now expanded on this paragraph.

Comment 5

Pg. 10, paragraph beginning on line 44 – Please modify this paragraph to present a more balanced description of why tourniquet use remains so prevalent. Please cite the purported benefits of

increased intraoperative visualization, and that intraoperative blood loss results are mixed, as multiple studies have found reduced blood loss.

Authors' response

Thank you for your comment. The previous study findings by Zhang et al. (2014) have now been superseded by the Cochrane review findings by Ahmed et al. (2020) and we have updated the citation in the text. The latter study includes a meta-analysis of all RCTs on this topic and has pooled results showing a statistically significant difference in intraoperative blood loss however no statistically significant difference in overall blood loss, change in haemoglobin values, and blood transfusion events in patients undergoing surgery with versus without a tourniquet. We have included further details regarding these findings in the text and discussed the purported effects of tourniquet use on improved surgical field of view.

Comment 6

Pg. 11 -12 – The limitations of this study are appropriately described. I look forward to reading a follow up to this study using the expanded data collected in updated versions of the NJR dataset.

Authors' response

Thank you for this feedback. We are currently in the process of devising a protocol for the follow up study.

VERSION 2 – REVIEW

REVIEWER	Sezgin , Erdem A Aksaray University, Orthopedics and Traumatology
REVIEW RETURNED	16-Mar-2021
GENERAL COMMENTS	Authors have successfully implemented the changes to in response to reviewers' comments. I believe it is suitable for publication. Thank you for the opportunity to review this work.